# Research on Pilot Control Strategy and Workload for Tilt-Rotor Aircraft Conversion Procedure

**Xufei Yan** [1] , **Ye Yuan** [2,*] **and Renliang Chen** [3]

1   Zhejiang Lab, Hangzhou 311100, China; yanxufei@zhejianglab.com
2   Department of Aerospace Engineering, Swansea University, Swansea SA1 8EN, UK
3   National Key Laboratory of Rotorcraft Aeromechanics, Nanjing University of Aeronautics & Astronautics, Nanjing 210016, China; crlae@nuaa.edu.cn
*   Correspondence: ye.yuan@swansea.ac.uk

**Abstract:** This paper studies the pilot control strategy and workload of a tilt-rotor aircraft dynamic conversion procedure between helicopter mode and fixed-wing mode. A nonlinear flight dynamics model of tilt-rotor aircraft with full flight modes is established. On this basis, a nonlinear optimal control model of dynamic conversion is constructed, considering factors such as conversion corridor limitations, pilot control, flight attitude, engine rated power, and wing stall effects. To assess pilot workload, an analytical method based on wavelet transform is proposed, which examines the mapping relationship between pilot control input amplitude, constituent frequencies, and control tasks. By integrating the nonlinear optimal control model and the pilot workload evaluation method, an analysis of the pilot control strategy and workload during the conversion procedure is conducted, leading to the identification of strategies to reduce pilot workload. The results indicate that incorporating the item of pilot workload in the performance index results in a notable reduction in the magnitude of collective stick inputs and longitudinal stick inputs. Moreover, it facilitates smoother adjustments in altitude and pitch attitude. Additionally, the conversion of the engine nacelle can be achieved at a lower and constant angular velocity. In summary, the conversion and reconversion procedures are estimated to have a low workload (level 1~2), with relatively simple and easy manipulation for the pilot.

**Keywords:** tilt rotor; conversion; optimal control model; pilot workload; wavelet transform

## 1. Introduction

Atilt-rotor aircraft combines the vertical takeoff and landing capabilities of a helicopter, and the high speed and range capabilities of a fixed-wing aircraft. It consists of three flight modes: helicopter mode, fixed-wing aircraft mode, and conversion mode. Conversion between helicopter mode and fixed-wing aircraft mode is completed by tilting the nacelle of the engine, as shown in Figure 1 [1].

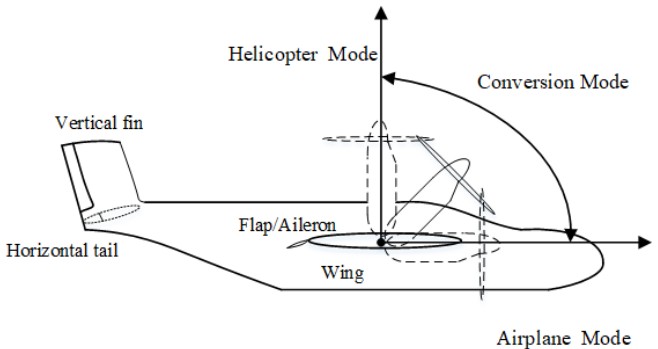

**Figure 1.** XV-15 tilt-rotor aircraft conversion flight mode.

The control process during conversion is complicated because of the cooperation between the lift and thrust, complex unsteady aerodynamic effects, body motion, and inertial coupling, as well as the control transition between helicopter mode and fixed-wing mode [2–4]. In the conversion procedure, the pilot not only has to focus on cockpit control but also pay attention to the tilting of the nacelles, which significantly increases the pilot workload [5,6].

Several studies have investigated the pilot control strategy during the conversion procedures of tilt-rotor aircraft between helicopter mode and fixed-wing mode [5–7]. Righetti conducted simulations of conversion maneuvers in order to minimize pilot workload [7]. However, the optimization results may be limited by a predetermined conversion path and nacelle tilting schedule, potentially leading to suboptimal outcomes. Yan transformed the pilot control strategy optimization problem during the conversion procedure into a nonlinear optimal control problem (NOCP) to relieve the pilot workload [5]. By studying the influence of different conversion paths on pitch attitude, altitude, and the power required during the conversion procedure, Yeo sought a better conversion path to improve flight attitude and mitigate altitude variations [6]. The aforementioned studies proposed control methods aimed at reducing the pilot workload through research and optimization of pilot control strategies based on high-precision flight dynamics models. However, these studies primarily focused on qualitative analysis of smooth pilot control, without further quantification and evaluation of the associated pilot workload. Therefore, it is necessary to conduct additional research to quantify and analyze the pilot workload in order to provide a comprehensive understanding of the topic.

Pilot workload is a hypothetical construct that represents the cost incurred by a pilot to achieve a particular flight task. Although many factors affect pilot workload, including the flight environment, pilot skills, perception, experience, etc., the level of the pilot workload is often estimated based on the time histories of the pilot control inputs [2,8–13]. Intuitively, the factors that primarily affect pilot workload can be described as the amplitude and frequency for each pilot control input, as well as the complexity of the cooperation between them. The limits of inherent pilot characteristics and high-frequency and large-scale operations, as well as the excessively complicated cooperation between the pilot control inputs, obviously increase the pilot workload. In other words, low-frequency and small-amplitude operations with simple coupling of control inputs can help to reduce the pilot workload.

In recent years, Padfield [8] introduced a metric that combines time-domain and frequency-domain measures to understand the relationship between a pilot's subjective assessment (Handling Quality Rating, HQR), measured control activity (pilot control stick input), and task performance, as shown in Figure 2. Lampton and Klyde [9,10] utilized wavelet analysis to calculate time-varying cutoff frequencies represented by time-frequency scale maps to quantify pilot control workload. Their research demonstrated a likely correlation between these frequencies and pilot HQR ratings based on flight tests, as shown in Figure 3. Tritschler and Mello [11–13] employed various pilot control action analysis methods to examine pilot control actions. They found that relying solely on cutoff frequency to determine pilot workload can be misleading. Conversely, by utilizing multifrequency component analysis, they identified major frequency components that closely aligned with pilots' descriptions and provided additional information on control frequencies and energy, as shown in Figure 4. These findings led to the proposal of a novel measure for quantifying pilot workload in both fixed-wing and rotary-wing aircraft. This measure, which was derived from wavelet analysis of pilot control activity, exhibited a strong correlation with pilot HQRs.

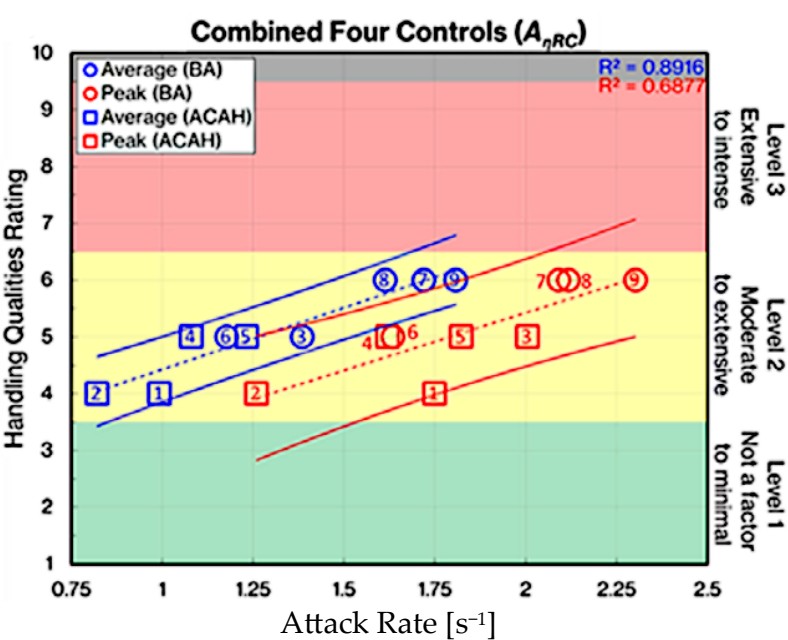

**Figure 2.** HQR vs. peak and average $A_{\eta RC}$ for combined controls [8]. $A_{\eta RC}$ is an adaptive metric calculated based on helicopter pilot control stick input data. Numbers: Case number, lines: $A_{\eta RC}$, dotted lines: best fit straight-line between the HQRs and $A_{\eta RC}$.

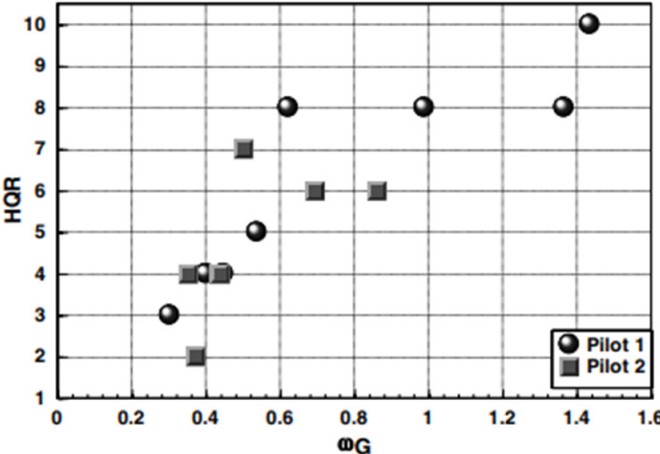

**Figure 3.** Handling quality ratings as a function of pilot lateral stick input power frequency [9].

| Frequency Range | Pilot Control Strategy/Task |
|---|---|
| 0.25–0.8 rad/sec (0.04–0.13 Hz) | Typical open-loop control associated with trimming and flight path modulation |
| 0.8–2.0 rad/sec (0.13–0.32 Hz) | Typical closed-loop control associated with transport aircraft maneuvering |
| 2.0–4.0 rad/sec (0.32–0.64 Hz) | Higher-gain closed-loop control associated with increased task urgency or handling issues with the aircraft, such as PIO |
| 4.0–10.0 rad/sec (0.64–1.59 Hz) | Very high-gain closed-loop control, almost certainly associated with control difficulties |

**Figure 4.** Proposed frequency ranges for control tasks [11,12], PIO: pilot-induced oscillation (PIO).

According to the research described above, pilot control action analysis methods based on time-frequency representations (TFRs) can more accurately quantify and assess pilot control workload. However, there is currently a lack of established metrics for accurate quantification of pilot workload in research on tilt-rotor aircraft conversion control strategy optimization.

In summary, many researchers are currently exploring control strategies based on nonlinear flight dynamics models and incorporating optimal control theory into the investigation of pilot control strategy during conversion procedures. However, these studies have primarily focused on qualitative analysis of smooth pilot control, neglecting further quantification and evaluation of the associated pilot workload. Furthermore, the analysis method based on time-frequency representations (TFRs) can provide a more accurate quantification and evaluation of pilot control workload. However, this method is currently only used to analyze real pilot control data after piloted-in-the-loop simulations or flight tests comparing the pilot control workload with pilot HQR ratings for validation. It has not yet been applied in the initial numerical simulation process.

Based on these two conclusions, it is evident that the combination of optimal control theory and TFR analysis methods allows for the acquisition of an optimal pilot control strategy during the simulation stage of the tilt-rotor aircraft conversion process, along with quantifiable measurements of pilot workload. This integrated approach provides a valuable reference for subsequent real-time pilot-in-the-loop simulations and final flight tests. Consequently, this research methodology serves to broaden the application scope of optimal control theory and TFR analysis methods. By evaluating and quantifying the pilot's optimal control strategy and workload at an early stage, valuable insights can be gleaned for the optimization and refinement of control system design.

This paper assumes that the use of TFRs is equally applicable for quantifying and assessing pilot workload during tilt-rotor aircraft conversion procedure. The structure of this paper is as follows. In Section 2, a nonlinear rigid-body dynamics model of a XV-15 tilt-rotor aircraft is established and validated using flight data. In Section 3, a nonlinear optimal control model of dynamic conversion is formulated with a pilot model, performance index, path constraints, and boundary conditions. An efficient numerical solution method with good convergence is developed to obtain the trajectory and control strategy. In Section 4, we propose a pilot control workload evaluation method. This method utilizes wavelet transform to obtain dominant frequency components of pilot control inputs and quantifies pilot workload by mapping these frequency components to the pilot control strategy (as described in [11,12]), as well as its potential relationship with the HQR. Finally, in Sections 5 and 6, the pilot control strategy, flight state, and pilot workload during XV-15 tilt-rotor aircraft forward conversion and backward reconversion procedures are compared and discussed on the basis of different performance indices. Conclusions are presented in Section 7.

## 2. Flight Dynamics Model of XV-15 Tilt-Rotor Aircraft

### 2.1. Modelling

In this paper, a XV-15 tilt-rotor aircraft is utilized as a sample. The nonlinear flight dynamics model of the XV-15 tilt-rotor aircraft [1] consists of two rotor models, a wing-pylon model, a fuselage model, a horizontal stabilizer model, a vertical stabilizer model, and a mixed control system. A brief introduction to the model is provided below.

1. The rotor is an important component of a tilt-rotor aircraft, providing both lift and thrust. In helicopter mode and conversion mode, the rotor functions as the lifting surface, control surface, and propulsor. In fixed-wing mode, the rotor acts as a propeller for high-speed forward flight. The aerodynamic forces of the rotor are influenced by the flapping motion of the blade and rotor-induced velocity, as shown in Figure 5. These forces and moments are calculated using the blade element theory. Lift, drag, and moment are determined for each blade element using a table lookup method, which considers flow separation and transonic compressibility effects.

2. The Pitt–Peters dynamic inflow model with rotor wake distortion effect and side-by-side rotor effect is applied to simulate the dynamic characteristics of rotor inflow during tilting [14].

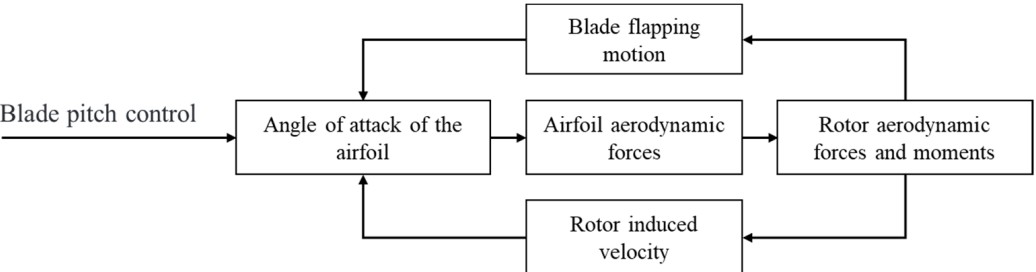

**Figure 5.** Tilt-rotor aircraft rotor aerodynamic characteristics.

3. The wing aerodynamic forces and moments due to rotor wake effects are calculated separately from the forces and moments generated by the freestream flow. The effect of rotor wake on the wing is a function of contracted wake radius, tilting angle, angle of attack, angle of sideslip, and dynamic pressure of the portion of the wing immersed in the wake (see details in the GTRS model in [15]), as shown in Figure 6, where $S_{wfs}$ represents the wing area in the free stream, $S_{wss}$ represents the wing area in the rotor slipstream, and $R_w$ represents the contracted wake radius.

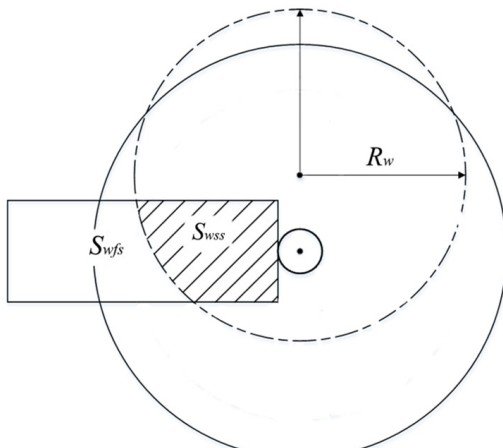

**Figure 6.** Diagram of aerodynamic influence of rotor wake on the wing.

4. The fuselage, wing–pylon assembly, horizontal tail, and vertical fins are modelled separately to account for the influence of rotor wake on the airframe aerodynamics (see details in the GTRS model in Ref. [15]).

Finally, according to the influence of nacelle tilting motion on rotor aerodynamics, the interference between aerodynamic components and inertial coupling during dynamic conversion, as well as the characteristics of control transition between helicopter mode and fixed-wing mode [15], a nonlinear flight dynamics model of tilt-rotor aircraft with full flight modes is established as follows

$$\dot{x}_b = f(x_b, u_b, t) \tag{1}$$

where $x_b = [x_B; x_F; x_I]$ represents the state vector, which contains the fuselage state ($x_B$), the rotor state ($x_F$) (left and right), and the inflow state ($x_I$); $u_b = \left[\delta_{col}; \delta_{lon}; \delta_{lat}; \delta_{ped}; i_n\right]$ is the control vector, where $\delta_{col}$ is the collective stick input, $\delta_{lat}$ is the lateral stick input, $\delta_{lon}$ is the longitudinal stick input, $\delta_{ped}$ is the pedal input, and the nacelle tilting angle ($i_n$) is applied to study the engine nacelle tilting law, which can be realized by the automatic tilting system; and $t$ is time.

$$\begin{cases} \boldsymbol{x}_B = [u, v, w, p, q, r, \phi, \theta, \psi, x, y, h]^{\mathrm{T}} \\ \boldsymbol{x}_F = \left[ \dot{a}_{0,R}, \dot{a}_{1,R}, \dot{b}_{1,R}, a_{0,R}, a_{1,R}, b_{1,R}, \dot{a}_{0,L}, \dot{a}_{1,L}, \dot{b}_{1,L}, a_{0,L}, a_{1,L}, b_{1,L} \right]^{\mathrm{T}} \\ \boldsymbol{x}_I = [v_{0,R}, v_{1s,R}, v_{1c,R}, v_{0,L}, v_{1s,L}, v_{1c,L}]^{\mathrm{T}} \end{cases} \tag{2}$$

where $u, v, w$ represent the linear velocities of the aircraft body axis system; $p, q, r$ correspond to the angular velocities of the body axis system; $\phi, \theta, \psi$ represent the roll, pitch, and yaw angles, respectively; $x, y, h$ denote the position of the aircraft in the Earth's axis system; $a_0, a_1, b_1$ denote the taper, rear, and side angles of the rotor disk, respectively; and $v_0, v_{1c}, v_{1s}$ represent the non-dimensional terms for rotor uniform inflow, first-order cosine inflow, and first-order sine inflow, respectively.

### 2.2. Validation

The flight test data of a XV-15 tilt-rotor aircraft (gross weight of 5897 kg) are utilized as a sample for steady-state analysis in three modes of flight. Figures 7 and 8 show a comparison between the predicted results and the flight test data [16]. Figure 8 shows the collective pitch angle ($\theta_0$), longitudinal stick displacement, vehicle pitch attitude, and total power required ($P_r$) against the flight test data for helicopter mode ($i_n = 90°$) with a flap/aileron setting of $40°/25°$, conversion mode ($i_n = 60°$ and $i_n = 30°$) with a flap/aileron setting of $20°/12.5°$, and fixed-wing aircraft mode ($i_n = 0°$) with a flap/aileron setting of $0°/0°$. $V_x$ represents the forward speed. The calculated steady-state variables are in agreement with the flight test data.

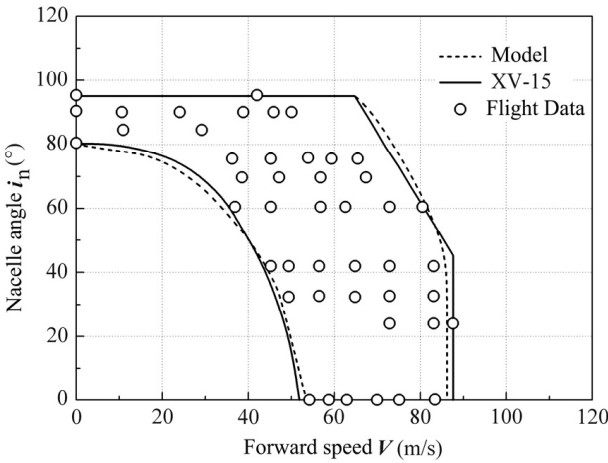

**Figure 7.** Conversion corridor of XV-15 tilt-rotor aircraft.

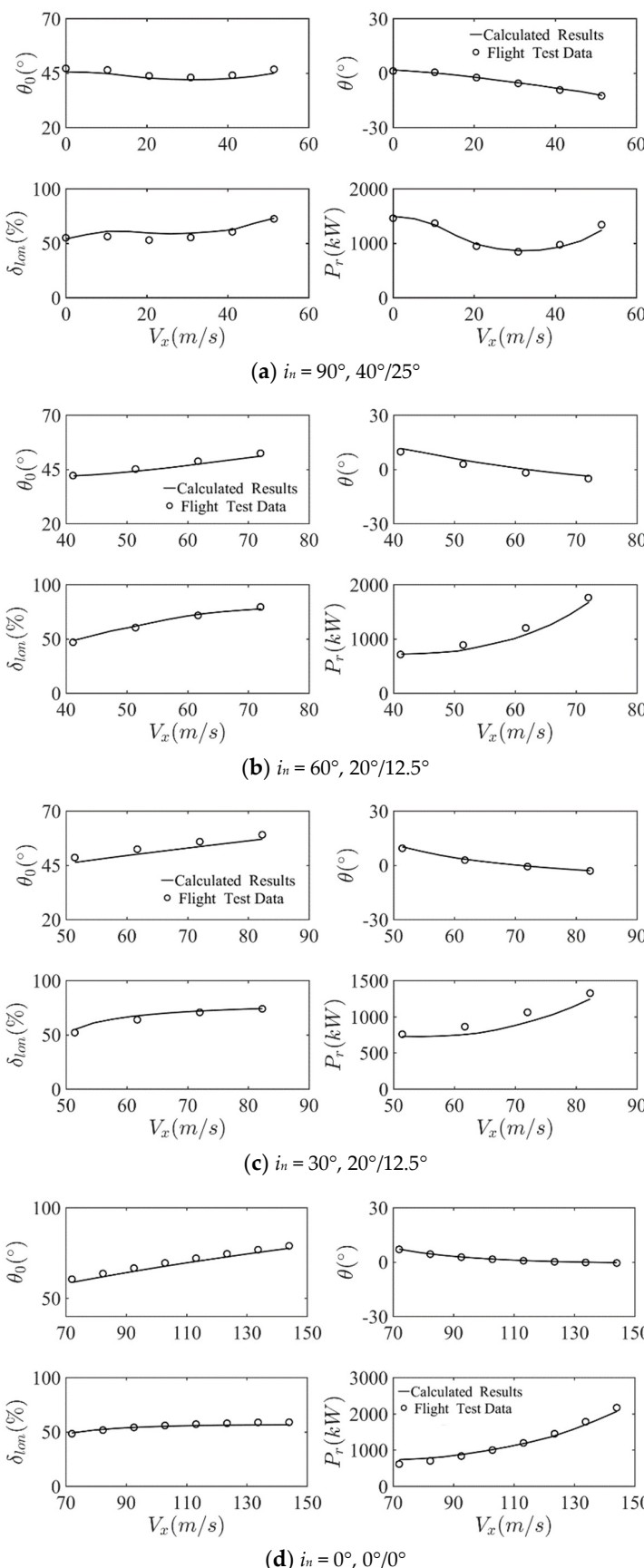

(**a**) $i_n = 90°$, 40°/25°

(**b**) $i_n = 60°$, 20°/12.5°

(**c**) $i_n = 30°$, 20°/12.5°

(**d**) $i_n = 0°$, 0°/0°

**Figure 8.** Comparison of the calculated trim results with flight data.

## 3. Formulation of Nonlinear Optimal Control Problem

The nonlinear optimal control model of dynamic conversion includes the differential equations constructed from the flight dynamics model of the tilt-rotor aircraft and the pilot model, as well as the cost function, path constraints, and boundary conditions formulated based on the flight missions, conversion corridor, and flight safety requirements.

### 3.1. Pilot Model

In order to reflect the pilot's control strategy during the dynamic conversion process of the tilt-rotor aircraft, a corresponding pilot model is first established and integrated into the optimal control model.

The primary task of the pilot is to perceive the current flight situation and determine appropriate control strategies based on the flight mission in order to manipulate the aircraft. Therefore, a pilot model generally needs to describe three characteristics of the pilot: perception system, control behavior, and neuromuscular system. Since the optimal control model developed in this paper can describe the desired flight mission through cost function, path constraints, and boundary conditions and obtain suitable control strategies through a numerical solution, it can be assumed that the pilot's control behavior is already incorporated into the optimal control model established in this paper [17], as shown in Figure 9, where $u_d$ represents the delayed command inputs, and $u_f$ represents the final control input to actuate the flight dynamics model. This section only requires the establishment of a model that describes the pilot's perception system and neuromuscular system.

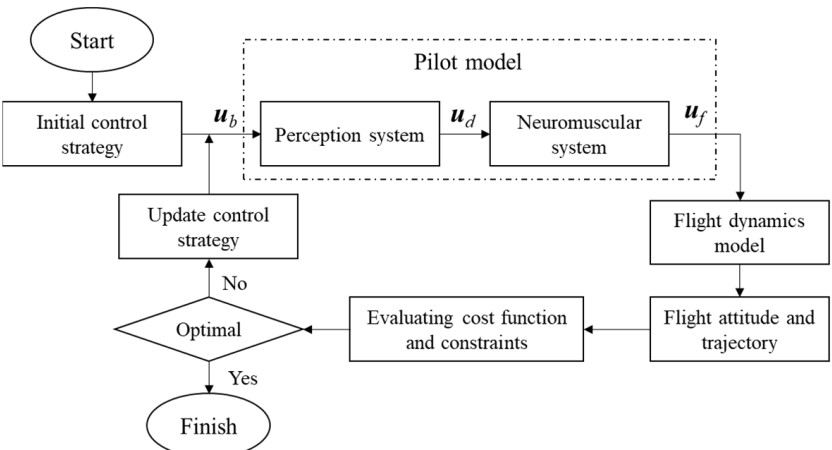

**Figure 9.** Schematic diagram of the optimal control model.

The perception system and neuromuscular system can be represented using the following transfer function:

$$H_p(s) = \frac{1 + T_L s}{1 + T_I s} \cdot \frac{e^{-\tau_p s}}{1 + T_N s} \tag{3}$$

where $T_L$ represents the feedforward time constant, which reflects the pilot's ability for anticipatory prediction, typically ranges between 0.1 s and 0.6 s, and is set to 0.25 s in this study based on relevant literature [16–18]. The inertia time constant ($T_I$) is used to represent the lag effect of the pilot's input displacement on the control stick, generally ranges between 0.2 s and 0.4 s, and is set to 0.25 s in this research. The inertia time constant ($T_N$) represents the lag caused by the pilot's neuromuscular response, typically ranges between 0.01 s and 0.2 s, and is set to 0.1 s in this study. The delay element ($e^{-\tau_p s}$) reflects the delayed response of the pilot. The delay time ($\tau_p$) is set based on the pilot's capabilities, usually ranges between 0.1 s and 0.25 s, and is set to 0.15 s in this study. Within the pilot's frequency range

(0.1–10 rad/s), the pure delay element can be approximated using a second-order Pade transfer function [17].

$$e^{-\tau_p s} \approx \frac{1 - \frac{1}{2}(\tau_p s) + \frac{1}{8}(\tau_p s)^2}{1 + \frac{1}{2}(\tau_p s) + \frac{1}{8}(\tau_p s)^2} \tag{4}$$

In each control input channel, the pilot model can be represented in the following state-space form:

$$\begin{cases} \dot{x}_p = Ax_p + Bu_b \\ u_f = Cx_p + Du_b \end{cases} \tag{5}$$

where $x_p$ represents the relevant state variables of the neuromuscular system. The elements of matrices $A$, $B$, $C$, and $D$ depend on the delay and filter parameters. The above equation can be integrated into the flight dynamics model to reflect the pilot's control actions during the numerical solution process of dynamic trajectory simulation. In order to account for the limits on the control rates of pilot control inputs, as well as the nacelle angle, and to avoid jump discontinuities arising in the time history of controls in the control optimization, time derivatives of $\delta_{col}, \delta_{lon}, \delta_{lat}, \delta_{ped}, i_n$ are applied as the control variables and denoted by $u_{col}, u_{lon}, u_{lat}, u_{ped}, u_n$, respectively. In the meantime, $\delta_{col}, \delta_{lon}, \delta_{lat}, \delta_{ped}, i_n$ are regarded as the state variables. Therefore, the augmented flight dynamics model can be expressed as follows:

$$\dot{x}_a = f_a(x_a, u_a, t) \tag{6}$$

where

$$\begin{cases} x_a = \left[x_B; x_F; x_I; u_b; x_p\right] \\ u_a = \left[u_{col}, u_{lon}, u_{lat}, u_{ped}, u_n\right]^{\mathrm{T}} \end{cases} \tag{7}$$

*3.2. Nonlinear Optimal Control Problem*

Based on the flight dynamics model and the characteristics of conversion between helicopter mode and fixed-wing aircraft mode, the tilt-rotor aircraft conversion procedure is formulated as a nonlinear optimal control problem (NOCP), which can be expressed as follows [5,19].

(1) **Differential equations** basically correspond to the augmented flight dynamics model (6). The tilt-rotor aircraft has a longitudinally symmetrical configuration; thus, the conversion procedure takes place in the longitudinal plane under no-crosswind conditions. In order to improve the calculation efficiency of the nonlinear optimal control method, this paper assumes that the state variables and control variables related to lateral and heading motion in the flight dynamics model remain in the initial state and do not participate in the numerical calculation of dynamic conversion.

(2) **Optimal variables** include differential state variables ($x_a$), control variables ($u_a$), and the free final time ($t_f$) (with the initial time ($t_0$) set to 0).

(3) **The cost function** of the NOCP is the performance index of the whole conversion procedure, which needs to consider the influence of multiple factors, such as the time of dynamic conversion, flight safety, feasibility, pilot workload, etc. Hence, the cost function ($J$) can be formulated as the following general expression:

$$\min_{u} J = \phi\left(x(t_0), t_0, x(t_f), t_f\right) + \int_{t_0}^{t_f} L(x(t), u(t), t)dt \tag{8}$$

where $t_0$ is the fixed initial time. The first term of expression Equation (8) represents the initial and terminal-state performance indices, and the second term represents the state and control performance indices of the whole conversion procedure. The specific cost function is given in Sections 5 and 6. The NOCP can be successfully solved if the time history of the control vector ($u(t)$) that minimizes the cost function is found under the following constraints.

(4) **Constraints:** The constraint equations consist of initial boundary conditions, path constraints, and terminal constraints.

The initial boundary conditions are the current flight state of the aircraft. The terminal constraints are set as the target tilting angle and forward flying speed:

$$
\begin{cases}
i_n(t_f) = i_{nt} \\
u_n(t_f) = 0°/\mathrm{s} \\
\dot{x}_{t,\min} \le \dot{x}(t_f) \le \dot{x}_{t,\max} \\
\dot{h}_{t,\min} \le \dot{h}(t_f) \le \dot{h}_{t,\max}
\end{cases}
\tag{9}
$$

where $i_{nt}$ is the target engine nacelle tilting angle, $\dot{x}_t$ is the target forward speed, $\dot{h}_t$ is the target ascent rate. Specific values and additional items can be determined according to the requirements of the conversion flight mission.

The path constraints should be determined according to the boundary of the conversion corridor, as shown in Figure 7. In low-speed conversion, the lift provided by the wing is limited by the critical stall angle of attack. Therefore, the wing angle of attack ($\alpha_W$) is at the critical value in the lower conversion envelope. The corresponding path constraint is

$$
\alpha_{WC,\min} \le \alpha_W(t) \le \alpha_{WC,\max}
\tag{10}
$$

where the critical angle of attack ($\alpha_{WC,\min}$ and $\alpha_{WC,\max}$) can be obtained from wind tunnel data [15]. The maximum forward speed in the conversion procedure is limited by the compressibility of the advancing rotor blade, the stall effect of the retreating rotor blade, and the available power and dynamic stability of the rotor, among which the available power of the rotor is dominant. Therefore, the path constraint determined by the upper conversion envelope is:

$$
0 \le P_r(t) \le P_n
\tag{11}
$$

where $P_n$ represents the rated power output of the engine. In order to ensure flight safety during the conversion process, the speed corresponding to the engine nacelle tilting angle of $45°$ on the upper conversion envelope is taken as the stop speed, and the flight speed during the conversion process must not exceed the stop speed ($V_{stop}$).

$$
V_{\max} \le V_{stop}
\tag{12}
$$

The constraints of altitude, pitch attitude, and pilot control inputs should also be considered in the path constraints. Notice that in order to study the influence of pilot control strategy on height change during dynamic conversion, the height constraint is appropriately relaxed. The constraints of control rates are selected according to the maximum physical rate limits of the servo booster.

$$
\begin{cases}
\Delta h_{\min} \le \Delta h(t) \le \Delta h_{\max} \\
-10° \le \theta(t) \le 10° \\
-5°/\mathrm{s} \le q(t) \le 5°/\mathrm{s}
\end{cases}
\tag{13}
$$

$$
\begin{cases}
0 \le \delta_{col}(t), \delta_{lon}(t) \le 1, 0° \le i_n(t) \le 90° \\
-0.3/\mathrm{s} \le u_{col}(t), u_{lon}(t) \le 0.3/\mathrm{s} \\
-15°/\mathrm{s} \le u_n(t) \le 15°/\mathrm{s}
\end{cases}
\tag{14}
$$

### 3.3. Numerical Solution Techniques

The state and control variables of the NOCP for the XV-15 tilt-rotor aircraft conversion procedure are numerous, and the cost function, as well as the constraints, is very complicated. Therefore, the optimal solution needs to be solved numerically. To improve

computational efficiency and the rate of convergence in the numerical optimization, the optimal variables of the NOCP are normalized and scaled first as follows:

$$
\begin{cases}
(\bar{u}, \bar{v}, \bar{w}) = \frac{k_1}{\Omega_0 R}(u, v, w), \quad (\bar{p}, \bar{q}, \bar{r}) = \frac{k_2}{\Omega_0}(p, q, r) \\
\left(\bar{\dot{a}}_{0,LR}, \bar{\dot{a}}_{1,LR}, \bar{\dot{b}}_{1,LR}\right) = \frac{k_2}{\Omega_0}\left(\dot{a}_{0,LR}, \dot{a}_{1,LR}, \dot{b}_{1,LR}\right), \quad \left(\bar{x}, \bar{y}, \bar{h}\right) = \frac{k_3}{R}(x, y, h) \\
\bar{\Omega} = \frac{\Omega}{\Omega_0}, \quad \tau = k_4 \Omega_0 t, \quad \frac{d(\cdot)}{d\tau} = \frac{1}{k_4 \Omega_0}\frac{d(\cdot)}{dt}
\end{cases} \tag{15}
$$

where $\Omega_0$ is the standard main rotor rotational speed, and $k_1 \sim k_4$ are the constant scaling factors. In order to ensure that the normalized, scaled optimal variables are close to values of one, take $k_1 = k_2 = 100$, $k_3 = 1$, $k_4 = 0.01$. The governing equations of the normalized and scaled flight dynamics model can be expressed as:

$$
\frac{d\bar{x}}{d\tau} = f(\bar{x}, \bar{u}, \tau) \tag{16}
$$

At present, the most effective and flexible approach to solve NOCP is to convert the NOCP into a nonlinear programming (NLP) problem via a collocation approach, which can then be solved with a nonlinear programming algorithm. In this paper, a collection approach called direct multiple shooting [20] is applied to transcribe the NOCP directly into a discrete NLP by breaking the states and controls of the continuous conversion procedure into shorter time segments. This approach is typically used in the applications of high complexity and/or with a large number of degrees of freedom [21]. The fundamental idea of a direct multiple-shooting approach is shown in Figure 10.

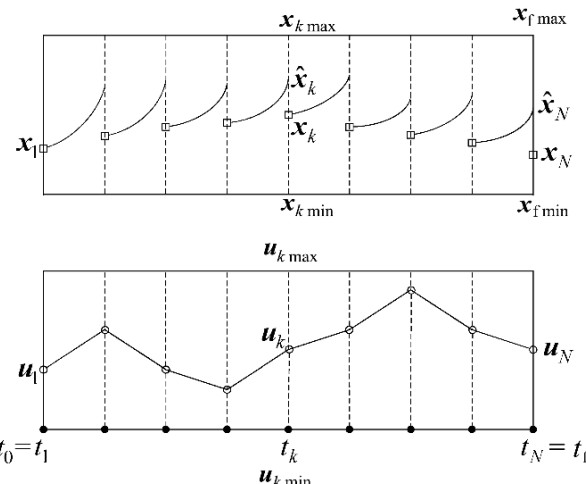

**Figure 10.** Direct multiple-shooting approach.

As shown in Figure 10, the solution time interval $[\tau_0, \tau_f]$ of the NOCP is divided into $N-1$ equal time segments. At the $k$-th time interval, we can integrate the differential equations from $\tau_k$ to the end of the segment at $\tau_{k+1}$ using the time-stepping approach with piecewise linear interpolation of $\bar{u}_k$ and $\bar{u}_{k+1}$, which helps to decrease the computational cost of finite differencing by increasing the problem sparsity. The multiple shooting segments are used to stabilize the integration of the vehicle equations of motion. This method guarantees that the discretized model is as close as possible to the original nonlinear model.

The result of this integration is denoted by $\hat{x}_k$; thus, the shooting of segment $k$ can be represented by

$$
\bar{x}_{k+1} - \hat{x}_{k+1} = 0, k = 1, \cdots, N-1 \tag{17}
$$

where

$$\hat{\bar{x}}_{k+1} = \bar{x}_k + \int_{\tau_k}^{\tau_{k+1}} f(\bar{x}, \bar{u}, \tau) d\tau. \tag{18}$$

The cost function of the NOCP is replaced by the performance index of the NLP in the same way:

$$\min J = \phi\left(\bar{x}(\tau_1), \tau_1, \bar{x}(\tau_N), \tau_N\right) + \sum_{k=1}^{N-1} \int_{\tau_k}^{\tau_{k+1}} L\left(\bar{x}(\tau), \bar{u}(\tau), \tau\right) d\tau. \tag{19}$$

The constraints are enforced on the corresponding time nodes. The nonlinear programming problem can be effectively solved using the SQP (series quadratic programming) algorithm [22] to obtain the approximate solution of the original NOCP. To improve the accuracy of the optimal solution, the optimal state variables ($x(t)$) are calculated by integrating the flight dynamics model (6) from $t_0$ to $t_f$ with piecewise linear interpolation of ($u_1, u_2, \ldots, u_k, \ldots, u_N$). In the sensitivity analysis, we observed that when the number of discrete points ($N$) exceeds 30, the numerical simulation results reach a steady state, and the computational efficiency decreases rapidly. Therefore, for this study, a value of 30 is chosen for the number of discrete points ($N$).

## 4. Workload Evaluation Method Based on Wavelet Analysis

Due to the limitations in metrics such as pilot aggressiveness and pilot cutoff frequency [11,12], which fail to directly capture the temporal variations in control actions, several new methods based on time-frequency domain representation have been proposed. Among these methods, the most prominent are short-time Fourier transform (STFT) and the recently developed wavelet analysis. Because STFT cannot simultaneously consider the needs of frequency and time resolution, the wavelet analysis method, with good time-frequency resolution, is more widely used in engineering [13]. This paper primarily focuses on studying pilot control actions using wavelet analysis and attempts to evaluate and analyze pilot control workload based on this approach.

Compared to power spectral density (PSD), time-frequency domain representation can provide insights into the distribution of signal energy in both the frequency spectrum and the time domain. The wavelet analysis method utilizes finite-length bandpass filters with a length of $g(t)$ equal to $n_c/\omega_c$ seconds, where $n_c$ represents the number of cycles. Each point in the wavelet analysis result ($G_{\delta\delta}(\omega, t)$) can be interpreted as the weighted power of the input signal ($\delta_x(t)$) at frequency $\omega_c$ within a time window of length $n_c/\omega_c$. The "weight" is determined by the wavelet function, which is defined as [23].

$$\int_{-\infty}^{\infty} \psi(t) dt = 0 \tag{20}$$

The choice of wavelet function is crucial in wavelet transformation, as different wavelet functions have significant waveform differences. Therefore, selecting an appropriate wavelet function is of great importance. In this section, we analyze the characteristics of wavelet functions and consider specific applications to select a suitable wavelet function. In signal recognition applications, the selection of wavelet functions can be based on the following characteristics [24]:

1. **Support set**: Wavelet functions can be divided into compactly supported and non-compactly supported based on their support length. A higher level of compact support indicates more concentrated energy, while non-compactly supported wavelets may result in energy loss in the decomposed signal, leading to increased recognition errors. Therefore, when recognizing a pilot's manipulation actions, wavelet functions with compact support are generally preferred.

2. **Orthogonality** refers to the property of orthogonality between the low-frequency and high-frequency components during wavelet function analysis. Orthogonality is benefi-

cial for the reconstruction of wavelet coefficients and is commonly used in image signal processing.

3. **Regularity** describes the smoothness level of a function. Wavelet functions with good regularity help improve the fitting performance between the wavelet basis and the signal, accurately describing the pilot's manipulation characteristics.

4. **Vanishing moments** indicate the concentration of energy after wavelet transformation. A higher order of vanishing moments filters the high-frequency components more effectively, indicating stronger denoising capabilities of the wavelet transformation. However, if the order of vanishing moments is too large, useful high-frequency components in the signal may be filtered out.

Considering the characteristics mentioned above and referring to related literature, the Daubechies (db) wavelet functions proposed by Ingrid Daubechies satisfy the requirements for analysis of a pilot's manipulation actions. The Daubechies wavelet functions exhibit orthogonality, compact support, good regularity, and suitable orders of vanishing moments [25]. Therefore, in this paper, we adopt the Daubechies wavelet functions for wavelet analysis of a pilot's manipulation actions.

Once the wavelet function is determined, the pilot's control time history (instantaneous varying signal) can be subjected to wavelet transformation. The wavelet family can be scaled by a scaling factor ($s$) and translated by a parameter ($u$). The scaling factor corresponds to frequency, while the translation parameter corresponds to time. Therefore, the wavelet function and its Fourier transform can be expressed as follows:

$$\begin{cases} \psi_{u,s}(t) = \frac{1}{\sqrt{s}}\psi\left(\frac{t-u}{s}\right) \\ \hat{\psi}_{u,s}(f) = e^{-i2\pi fu}\sqrt{s}\hat{\psi}(sf) \end{cases}. \tag{21}$$

The wavelet transform of the time-varying signal can be expressed as follows:

$$W_y(u,s) = \int_{-\infty}^{\infty} y(t)\psi_{u,s}^*(t)dt = \int_{-\infty}^{\infty} \hat{y}(f)\hat{\psi}^*_{u,s}(f)df \tag{22}$$

where $W_y(u,s)$ is referred to as the wavelet coefficient, and * denotes the conjugate relationship.

According to above equation, when the center frequency of the wavelet is close to certain frequency components in the original signal, the wavelet coefficient takes on maximum values. Therefore, the wavelet function can be seen as a bandpass filter that only allows signals with frequencies close to the center frequency of the wavelet to pass through. During the wavelet transformation process, a series of center frequencies can be obtained by scaling factors, while different time-frequency information of the signal can be detected by translation coefficients. This provides information about the frequencies and amplitudes contained in the signal at different time points. Hence, wavelet analysis can identify multiple dominant frequency components in the pilot's control input. To enhance the numerical efficiency of wavelet analysis, a sampling frequency of 20 Hz is chosen, given that the pilot's control frequencies can be identified up to 10.0 rad/s (1.6 Hz). After sampling, the initial stick deflection displacement is subtracted, since the pilot does not apply any control deflection in the initial state (energy value of 0 for the initial signal). For the wavelet transform, the widely used db3 wavelet [25] function is selected.

We identified multiple dominant frequency components in the pilot's control input following the approach proposed by Tritschler and O'Connor in [11,12]:

1. To determine the frequencies corresponding to each moment during maneuvering, we identified peaks in the time-frequency representation (TFR) at each time instant. For instance, Figure 11a presents a vertical slice of the two-dimensional spectrogram at 49 s, showing six prominent peaks. It is important to note that only local extrema exceeding a certain threshold in the spectrogram magnitude are considered as peaks. To date, no specific mathematical criterion has been proposed to select this threshold. In our study, we set it at 8% of the maximum value in the continuous-time TFR according to Tritschler and O'Connor's work.

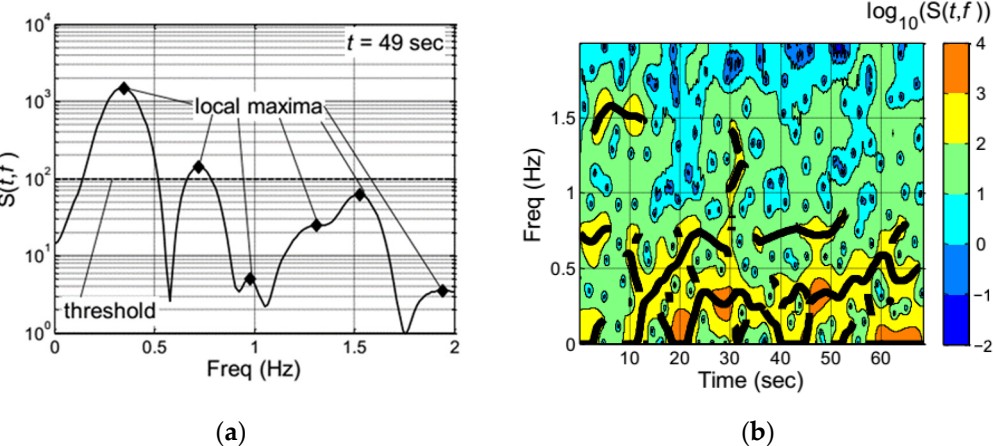

**Figure 11.** Determination of "peaks" and "ridgelines" in the TFR of pilot lateral cyclic control input for a hover MTE (mission task element). (**a**) Peaks at a given instant in the TFR, (**b**) Ridgelines in the continuous TFR, black lines: ridgelines.

2. The subsequent step in the multifrequency approach involves the mathematical identification of consecutive local maxima in time that signify continuous features in the TFR. Specifically, we aim to identify pairs of local maxima occurring at (or close to) the same frequency. In our study, we employed a threshold of 0.02 Hz to determine the continuity between two local maxima at consecutive time instants. This step essentially seeks to mathematically pinpoint the "peaks" and "ridgelines" in the topographical representation of the TFR, as illustrated in Figure 11b.

According to the research conducted by Tritschler and O'Connor, these frequency components may correspond to various flight tasks or control strategies employed by the pilot, as summarized in Figure 4 [11,12]. It is important to note that the frequency ranges presented in Figure 4 (along with their associated control task descriptions) were derived specifically from fixed-wing aircraft control inputs and may not directly apply to the control inputs of tilt-rotor aircraft. Nonetheless, there is still reason to believe that there could be some connection between control action frequencies and the flight tasks.

By observing the control strategy descriptions corresponding to component ranges of different frequencies in Figure 4 and considering the pilot's handling qualities rating (HQR) based on the Cooper–Harper scale, as shown in Figure 12 [10], a correspondence can be found between the two.

Based on the pilot control descriptions presented in Figures 4 and 12, it can be hypothesized that the dominant frequency range of 0.25–0.8 rad/s is likely associated with HQR level 1 (1~3), the range of 0.8–2.0 rad/s corresponds to HQR level 2 (4~6), the range of 2.0–4.0 rad/s is indicative of HQR level 3 (7~9), and the flight task description aligning with the range of 4.0–10.0 rad/s corresponds to an HQR rating of 10, as outlined in Table 1.

Therefore, the pilot control workload assessment method can be outlined as follows. First, the dominant frequency components of pilot control actions are extracted using wavelet transform. Next, peaks in the time-frequency representation (TFR) are identified at each time instant, enabling the identification of "peaks" and "ridgelines" in the TFR's topographical depiction. Subsequently, the potential mapping relationship between dominant frequency components corresponding control strategy descriptions and the Cooper–Harper HQR scale are utilized to predict the pilot's control workload level and potential rating range. This method provides a detailed description of the actual amplitude and frequency of pilot control inputs while also offering a quantitative measure through the simple HQR rating. Such an approach allows for a more comprehensive analysis of the pilot's control characteristics, aiding in the development of control strategies that are both more reasonable and impose a lower control workload.

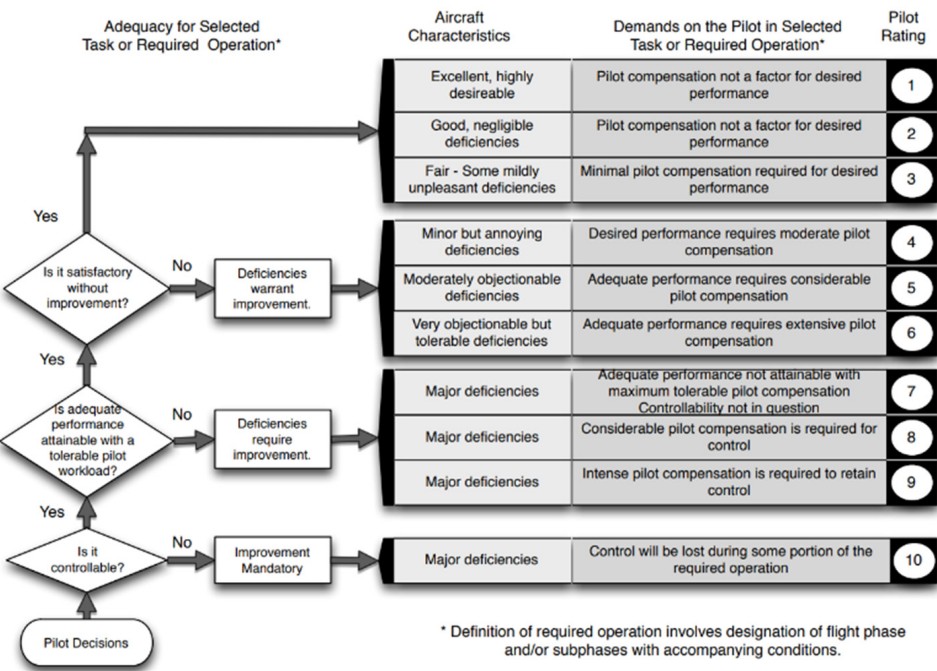

**Figure 12.** Cooper–Harper handling qualities rating scale.

**Table 1.** Potential mapping relationship between dominant frequency components, corresponding control strategy descriptions, and the Cooper–Harper HQR scale.

| Dominant Frequency Range | Pilot Control Strategy/Task | HQR Scale |
|---|---|---|
| 0.25–0.8 rad/s (0.04–0.13 Hz) | Typical open-loop control associated with trimming and flight path modulation | Level 1 (1~3) |
| 0.8–2.0 rad/s (0.13–0.32 Hz) | Typical closed-loop control associated with transport aircraft maneuvering | Level 2 (4~6) |
| 2.0–4.0 rad/s (0.32–0.64 Hz) | Higher-gain, closed-loop control associated with increased task urgency or handling issues with the aircraft, such as PIO | Level 3 (7~9) |
| 4.0–10.0 rad/s (0.64–1.59 Hz) | Very high-gain, closed-loop control almost certainly associated with control difficulties | 10 |

## 5. Tilt-Rotor Aircraft Forward Conversion Procedure

### 5.1. Task Description

In this section, we employ the optimal control method established in Section 3 and the pilot workload evaluation method developed in Section 4 to investigate the control strategy for forward conversion of tilt-rotor aircraft and evaluate the pilot workload. An XV-15 tilt-rotor aircraft is taken as an example.

Relevant literature and flight tests [5,16,22] indicate that the initial speed for forward conversion in helicopter mode is generally between 30.6 m/s and 41.7 m/s, corresponding to a low-power state in helicopter mode. In the final stage of the conversion, the forward flight speed range is typically limited to 61.7 m/s to 72 m/s, corresponding to a low-power state in fixed-wing mode. Additionally, the rotor speed is maintained at 589 r/min throughout, while the aileron/flap configuration remains at 40°/25°. Once in fixed-wing mode and stabilized, the pilot reduces the rotor speed to minimize vibration levels and switches the aileron/flap configuration to 0°/0° to enter high-speed, fixed-wing aircraft

mode. Therefore, the initial state of conversion is determined as follows: mass of 5897 kg, aft center of gravity, standard atmospheric conditions, no crosswind, steady-level forward flight speed of 35 m/s, and altitude of 100 m, with a 2 s wait. After conversion, the speed is maintained at 61.7 m/s to 72 m/s In steady-level flight, with the aileron/flap configuration remaining at 40°/25° throughout. The boundary condition can be specified as follows:

$$
\begin{cases}
i_n(t_f) = 0° \\
61.7\text{m/s} \leq \dot{x}(t_f) \leq 72\text{m/s}, \dot{h}(t_f) = 0\text{m/s} \\
q(t_f) = 0°/\text{s}, \dot{q}(t_f) = 0°/\text{s}^2 \\
\dot{u}(t_f) = 0\text{m/s}^2, \dot{w}(t_f) = 0\text{m/s}^2 \\
u_n(t_f) = 0°/\text{s}, u_{col}(t_f) = 0\%/\text{s}, u_{lon}(t_f) = 0\%/\text{s}
\end{cases}
. \tag{23}
$$

If a tilt-rotor aircraft encounters danger in conversion mode, it is unable to quickly maneuver to avoid it. Furthermore, it cannot quickly enter into autorotation when the engine fails in conversion mode. Therefore, the boundary condition is one of the most important indices to complete the conversion as soon as possible within the safety range. In this section, we consider the minimum time performance index as a benchmark and subsequently supplement it with the pilot workload item.

*5.2. Benchmark Performance Index*

The benchmark performance index of minimum time is set as

$$
\min J_{C1} = \tau_f - \tau_0. \tag{24}
$$

Figure 13 presents the computed results of the forward dynamic conversion control strategy under the benchmark performance index ($J_{C1}$).

As shown in Figure 13, the engine nacelle tilts at a maximum angular rate of 15°/s, taking approximately 6 s to tilt into fixed-wing aircraft mode. However, Figure 13b shows that the engine tilt angular rate experiences a jump between 0°/s and 15°/s at the beginning and end, resulting in a sudden increase in tilt angular acceleration. This is not only detrimental to the design of the tilt control system (instantaneous angular acceleration is too high) but also affects the pitch attitude of the aircraft, thereby impacting flight quality (large pitch attitude variation during conversion), as shown in Figure 13g,h. Additionally, the collective pitch stick undergoes a change of 27% in amplitude, with a rate ranging between −20%/s and 20%/s. The longitudinal stick changes by 66% in amplitude, with a rate jumping between −30%/s and 30%/s, indicating relatively aggressive pilot manipulation.

To further illustrate the level of aggressiveness in pilot manipulation under performance index $J_{C1}$, the workload evaluation method based on wavelet analysis is applied to the pilot's manipulation of the collective pitch stick and longitudinal stick, followed by an evaluation of the pilot's workload.

Figure 14 presents the wavelet analysis results of the pilot's manipulation actions during forward conversion under benchmark performance index $J_{C1}$. Figure 14a shows the wavelet transform result of the collective pitch stick. It can be observed that there is a high-energy input in the range of 0.2 rad/s to 2.0 rad/s, occurring from 2 s to 6 s. This indicates a significant amplitude of pilot input. After 12 s, there is also a small-amplitude manipulation around 0.5 rad/s, corresponding to the balancing process of the collective pitch stick. Furthermore, there are some lower-energy (small-amplitude), high-frequency components (exceeding 2 rad/s) present at 4 s, 6 s, and 10 s. These are caused by the continuous jumping of the collective pitch control rate within the constraint range. Figure 14b displays the wavelet transform result of the pilot manipulation of the longitudinal stick. Comparing it with Figure 13, it can be observed that the maximum energy input of the longitudinal stick is significantly greater than that of the collective pitch stick. This is due to the more drastic amplitude changes of the longitudinal stick, requiring the pilot to input greater energy for manipulation. The longitudinal stick exhibits a medium-energy input in the range of 0.2 rad/s to 1.0 rad/s from 2 s to 5 s, while a

high-energy input in the range of 0.1 rad/s to 1.5 rad/s occurs from 6 s to 10 s. These frequency components correspond to the pilot adjusting the flight attitude. Additionally, there are some lower-energy (small-amplitude), high-frequency components (exceeding 2 rad/s) present at around 3 s, 5 s, 8 s, and 10 s. These are also a result of the longitudinal stick control rate continuously jumping within the constraint range.

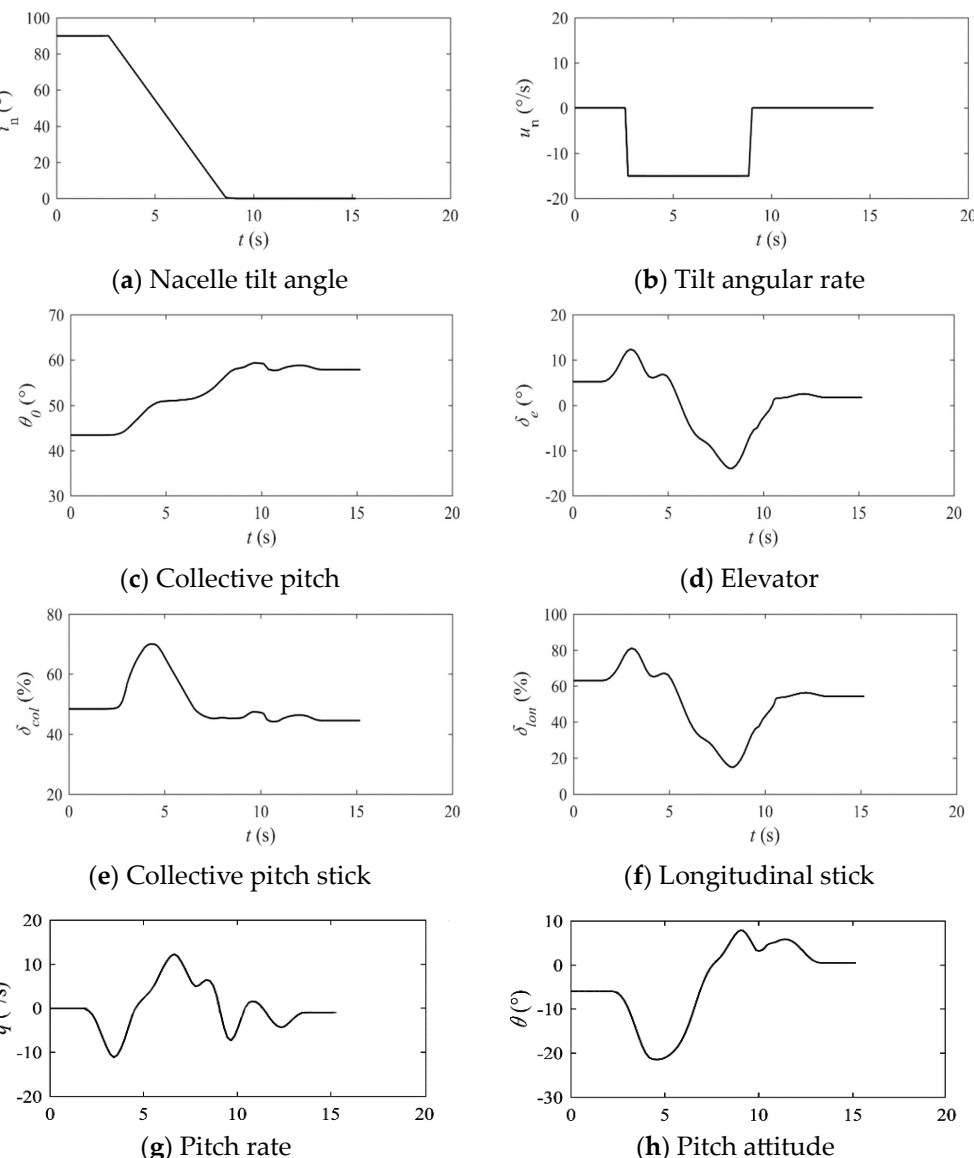

**Figure 13.** Forward conversion control strategy and attitude under benchmark performance index $J_{C1}$.

A comparison with Table 1 evidences that the low-frequency, high-energy density components in pilot manipulation primarily correspond to flight path adjustments and conventional maneuvering. On the other hand, high-frequency, low-energy density components pose challenges for flight tasks and may even induce pilot-induced oscillations. According to the pilot control workload assessment method proposed in Section 4 (Table 1), the benchmark performance index ($J_{C1}$) indicates a high control workload for forward dynamic conversion corresponding to HQR level 3 based on the dominant high-frequency components. This implies a significant control burden, making it difficult for the pilot to achieve a safe and feasible forward conversion process. Therefore, it is necessary to consider the pilot's workload in the performance index to prevent high-frequency jumps in the manipulation rates of the collective pitch stick and longitudinal stick.

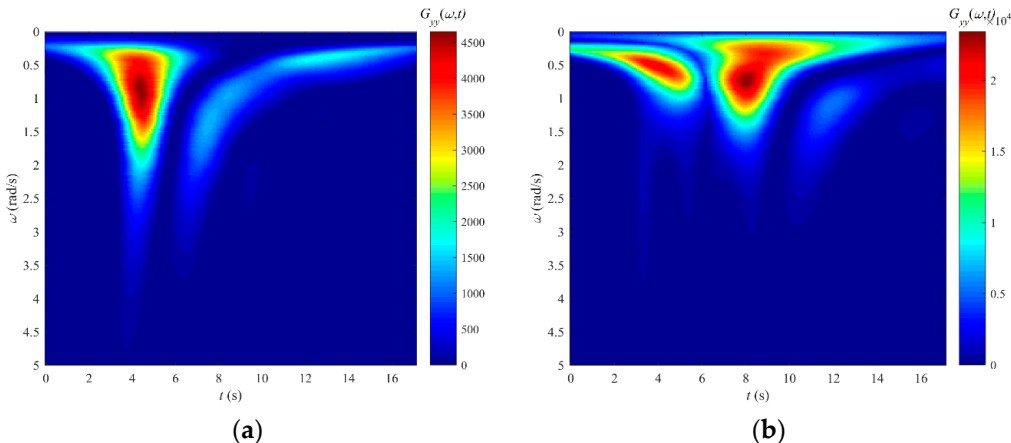

**Figure 14.** Wavelet analysis results of pilot manipulation under benchmark performance index $J_{C1}$. (**a**) Collective pitch stick wavelet transform result (unit: %2/(rad/s)). (**b**) Longitudinal stick wavelet transform result (unit: %2/(rad/s)).

### 5.3. Weighting of Pilot Workload in Performance Index

To incorporate the pilot workload into the performance index, this paper refers to Carlo's research findings and introduces a weighting factor for the objective pilot control rate component. Carlo's study demonstrates that as the pilot's control rate decreases, the variations in their control input become smoother and contain fewer high-frequency components, which is beneficial for reducing the pilot's control workload [17].

$$
\min J_{C2} = w_t \left( \tau_f - \tau_0 \right) + \frac{w_p}{\tau_f - \tau_0} \int_{\tau_0}^{\tau_f} \left[ w_{col} \cdot u_{col}^2 / u_{col,\max}^2 + w_{lon} \cdot u_{lon}^2 / u_{lon,\max}^2 + w_n \cdot u_n^2 / u_{n,\max}^2 \right] d\tau
\tag{25}
$$

where the first term of $J_{C2}$ represents the minimum time target item; the second term represents the minimum control rate target item of the whole optimal landing procedure; $u_{col,\max}$, $u_{lon,\max}$, and $u_{n,\max}$ are the maximum values of the control rates; $w_t$ represents the time weighting coefficient; and $w_p$ represents the pilot control workload weighting coefficient. Weighting coefficients are assigned to the control rates in order to assess their relative importance; thus, $w_{col}$, $w_{lon}$, and $w_n$ represent the weighting coefficients for the corresponding control rates.

First, the weighting coefficients for the control rates are established. Figure 13b reveals that there are abrupt jumps between $0°/s$ and $15°/s$ between the initial and final stages, which are unfavorable for the design of the pitching control system and can impact flight quality. To address these issues and ensure a smooth transition of pitch rates, the weighting coefficient ($w_n$) is considered for engine nacelle control. Additionally, the pilot primarily focuses on controlling the collective stick and longitudinal stick during the pitching process. Consequently, the weighting coefficients ($w_{col}$ and $w_{lon}$) for these controls should be higher than $w_n$ for engine nacelle control. Furthermore, Figure 14 demonstrates that the frequency variation of the longitudinal stick control is more pronounced, necessitating a larger weighting coefficient ($w_{lon}$) to reduce the control workload. Based on the analysis, the following weighting coefficients are set for the control variables: $w_{lon} = 0.5$, $w_{col} = 0.35$, and $w_n = 0.15$.

The allocation of the pilot control workload weighting coefficient ($w_p$) and the time weighting coefficient ($w_t$) significantly impacts the control strategy for the forward conversion procedure. If the value of $w_p/w_t$ is too low, the time is shorter but at the expense of increased pilot control workload. Conversely, if $w_p/w_t$ is too high, the pilot control workload decreases, but the time needed increases. The attitude and altitude variations also vary under different values of $w_p/w_t$. To determine a reasonable allocation of weighting coefficients, in this section, we compare and analyze the pilot control strategy and flight

state variations in association with different values of $w_p/w_t$. Based on this analysis, the allocation of weighting coefficients for various parameters in the performance index ($J_{C2}$) can be determined. When analyzing the impact of $w_p/w_t$ on the control strategy, in this study, we carefully selected representative values. For analysis of the forward conversion procedure, significant values of 0, 1/9, 3/7, and 1/1 were chosen, as they effectively demonstrate the influence of $w_p/w_t$ on the control strategy.

Figure 15 shows the pilot control strategy and flight state under different $w_p/w_t$ values.

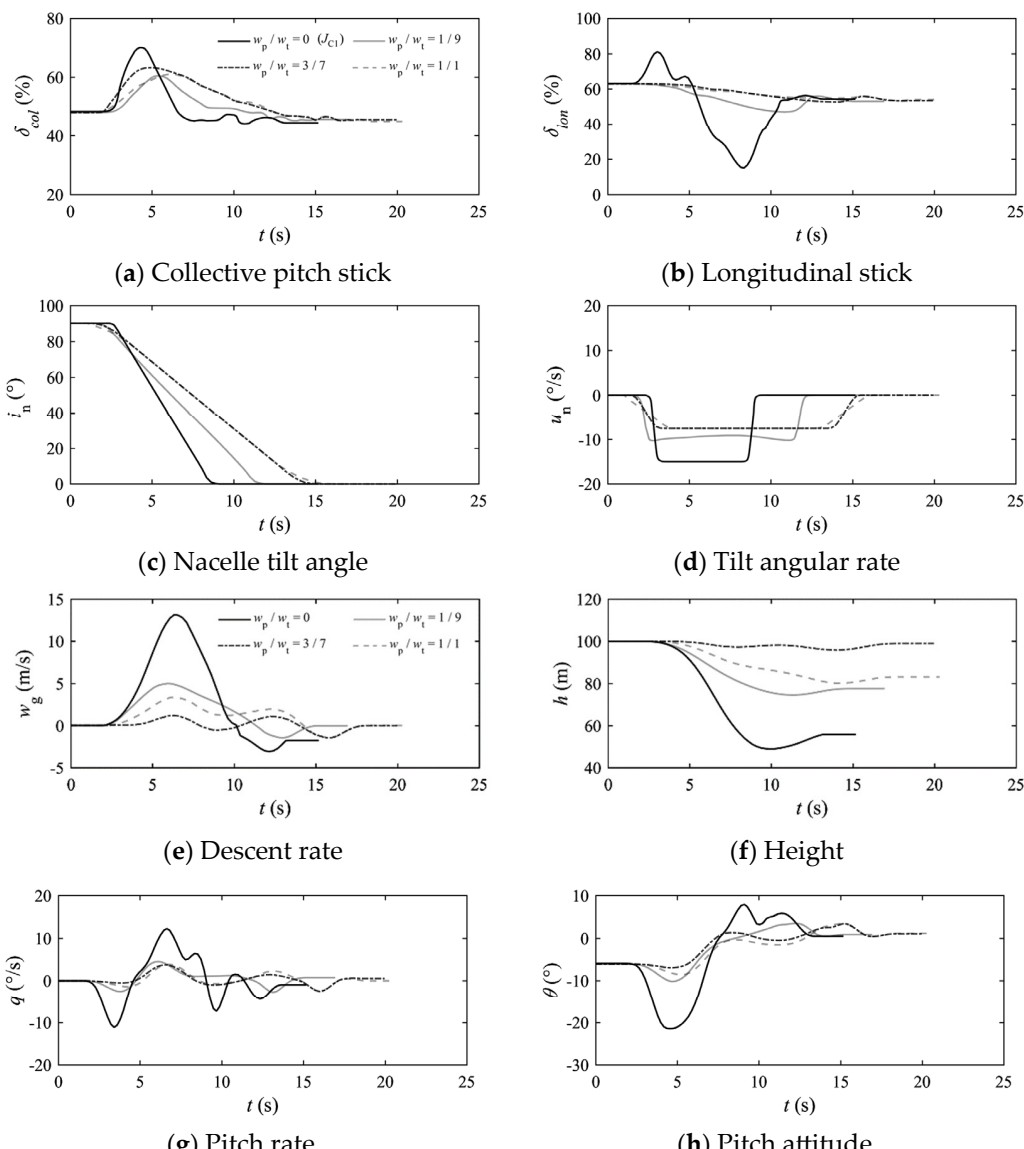

**Figure 15.** Forward conversion in association with different $w_p/w_t$ values under performance index $J_{C2}$.

As shown in Figure 15, when the performance index considers the weight coefficient of pilot workload ($w_p$), the displacements of controls are significantly reduced, and the height and pitch attitude change more gently, although the time is extended. In addition, $w_p$ should not be dominant; otherwise, overly smooth manipulation leads to a decrease in height. When the ratio of $w_p/w_t$ is 3/7 for the conversion procedure, pilot manipulation is smooth, the height change is small, and the tilting process of the engine nacelle can be

basically realized by the constant angular rate automatic tilting system (stabilized at 7.5°/s). Therefore, the performance index is set as

$$
\begin{aligned}
\min J_{C2} = {}& 0.7\left(\tau_f - \tau_0\right) + \\
& \frac{0.3}{\tau_f - \tau_0} \int_{\tau_0}^{\tau_f} \left[ w_{col} \cdot u_{col}^2 / u_{col,\max}^2 + w_{lon} \cdot u_{lon}^2 / u_{lon,\max}^2 + w_n \cdot u_n^2 / u_{n,\max}^2 \right] d\tau
\end{aligned}
\tag{26}
$$

Figure 16 presents the wavelet analysis results of the pilot's manipulation actions during forward conversion under benchmark performance index $J_{C2}$. Table 2 shows a comparison of wavelet analysis results presented in Figures 14 and 16.

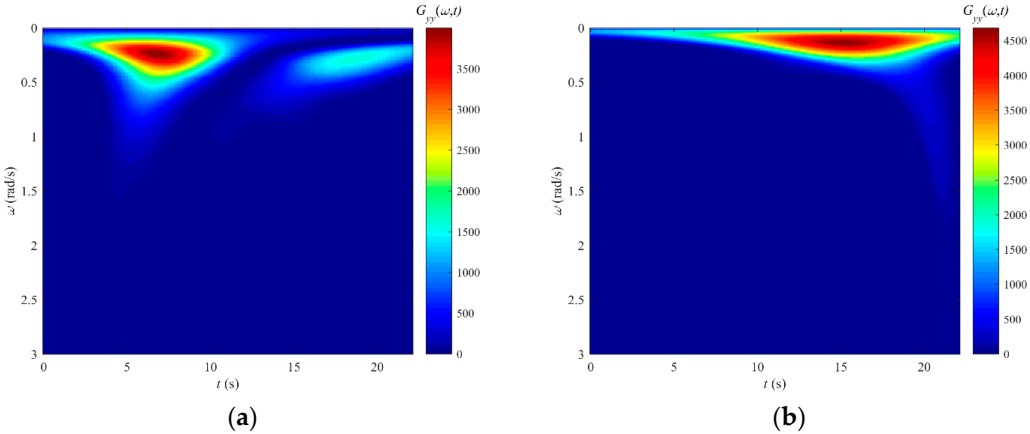

(**a**)  (**b**)

**Figure 16.** Wavelet analysis results of pilot manipulation under performance index $J_{C2}$. (**a**) Collective pitch stick wavelet transform result (unit: %²/(rad/s)). (**b**) Longitudinal stick wavelet transform result (unit: %²/(rad/s)).

**Table 2.** Comparison of wavelet analysis results: Figure 14 vs. Figure 16, forward conversion procedure.

| Control | Item | Figure 14 | Figure 16 |
|---|---|---|---|
| Collective pitch stick | Dominant frequency components | 0.2~2.0 rad/s, ~2.0 rad/s | 0.1~0.8 rad/s, <1.6 rad/s |
| | Maximum energy | 4500%²/(rad/s) | 3900%²/(rad/s) |
| Longitudinal stick | Dominant frequency components | 0.1~1.5 rad/s, ~2.0 rad/s | 0.1~0.8 rad/s, <1.8 rad/s |
| | Maximum energy | 23,000%²/(rad/s) | 4600%²/(rad/s) |

Figure 16a shows the wavelet transform result of the collective pitch stick. It can be observed that the overall energy distribution of the collective pitch stick is similar to that shown in Figure 14a but with a 15% decrease in maximum energy input, as shown in Table 2. The main energy inputs are concentrated around 0.1 rad/s to 0.8 rad/s, and the high-frequency components with low energy (small amplitudes) are also below 1.6 rad/s. This is because the performance metric ($J_{C2}$) considers the control rate of the collective pitch stick, preventing it from jumping abruptly. Figure 16b displays the wavelet transform results of the longitudinal stick. A comparison with the results presented in Figure 14b and Table 2 reveals a significant decrease (80%) in the maximum energy of the longitudinal rod, with the main energy inputs corresponding to frequencies below 0.3 rad/s, representing the pilot's adjustments of the flight attitude. The low-energy (small-amplitude), high-frequency components appearing after 20 s are also below 1.8 rad/s, corresponding to the final trim control process.

With reference to Table 1 presented in Section 4, it can be inferred that pilot manipulation involves trim and flight path adjustments (with most of the energy inputs

corresponding to frequency components below 0.8 rad/s). Only a small portion of the energy inputs corresponds to maneuvering actions seen in transport aircraft (frequency components ranging from 0.8 to 2.0 rad/s). Based on the mapping relationship between frequency components and workload proposed in this paper, it is likely that under performance metric $J_{C2}$ the pilot control workload falls between level 1 and level 2 of the HQR scale (rated as 3–4), indicating a low-workload forward conversion process with relatively simple and easy manipulation for the pilot.

## 6. Tilt-Rotor Aircraft Backward Reconversion Procedure

### 6.1. Task Description

In this section, we employ the optimal control method established in Section 3 and the pilot workload evaluation method developed in Section 4 to investigate the control strategy for backward reconversion of tilt-rotor aircraft and evaluate the pilot workload. An XV-15 tilt-rotor aircraft is taken as an example.

According to relevant literature and flight tests [5,16,22], the pilot initiates deceleration in high-speed, fixed-wing mode, simultaneously increasing the rotor speed to 589 r/min and switching the aileron/flap configuration to 40°/25° in preparation for the reconversion. At this stage, the forward flight speed range is generally between 61.7 m/s and 72 m/s (with a tilt angle of the engine nacelle of $i_n = 0°$), corresponding to a low-power state in fixed-wing mode. In the final stage of the reconversion, the forward flight speed range is typically restricted to 30.6 m/s to 41.7 m/s, corresponding to a low-power state in helicopter mode. The aileron/flap configuration and rotor speed remain the same as in the forward conversion process. Therefore, the initial state of reconversion is determined as follows: mass of 5897 kg, aft center of gravity, standard atmospheric conditions, no crosswind, steady-level forward flight speed of 65 m/s, and altitude of 100 m, with a 2 s wait. After backward reconversion, the speed is maintained in the range of 30.6 m/s to 41.7 m/s in steady-level flight, with the aileron/flap configuration remaining at 40°/25° throughout. The boundary condition can be specified as follows:

$$\begin{cases} i_n(t_f) = 90° \\ 30.6\,\text{m/s} \leq \dot{x}(t_f) \leq 41.7\,\text{m/s},\ \dot{h}(t_f) = 0\,\text{m/s} \\ q(t_f) = 0°/\text{s},\ \dot{q}(t_f) = 0°/\text{s}^2 \\ \dot{u}(t_f) = 0\,\text{m/s}^2,\ \dot{w}(t_f) = 0\,\text{m/s}^2 \\ u_n(t_f) = 0°/\text{s},\ u_{col}(t_f) = 0\%/\text{s},\ u_{lon}(t_f) = 0\%/\text{s} \end{cases} . \tag{27}$$

Similar to the research process for the forward conversion performance index, in this section, we first adopt the minimum time performance index as the primary benchmark, followed by supplementary analysis of the pilot workload item.

### 6.2. Benchmark Performance Index

The benchmark performance index of minimum time is set as

$$\min J_{R1} = \tau_f - \tau_0. \tag{28}$$

Figure 17 presents the computed results of the backward dynamic reconversion control strategy under benchmark performance index $J_{R1}$.

Figure 17 show that when considering only the time aspect of the performance index, the engine nacelle tilts at a maximum angular rate of 15°/s, requiring approximately 6 s to transition back to helicopter mode. However, as depicted in Figure 17b, there are noticeable jumps in the engine tilt angular rate between 0°/s and 15°/s between the initial and final stages, resulting in a sudden increase in tilt angular acceleration. Additionally, the collective pitch stick displacement exhibits a significant variation, reaching its minimum point (0%) at 8 s. The longitudinal stick displacement also demonstrates intense fluctuations, with the control rate fluctuating between −30%/s and 30%/s.

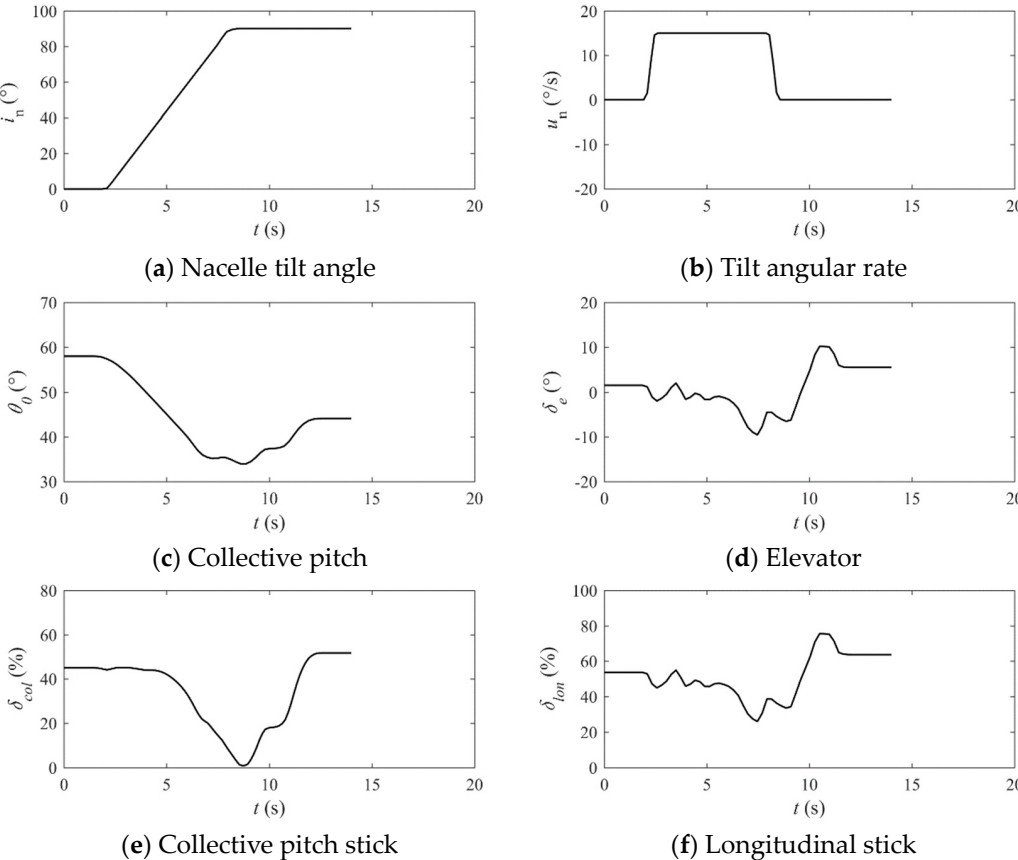

**Figure 17.** Backward reconversion control strategy under benchmark performance index $J_{R1}$.

To further illustrate the level of aggressiveness of pilot manipulation under performance index $J_{R1}$, the workload evaluation method based on wavelet analysis is applied to the pilot's manipulation of the collective pitch stick and longitudinal stick, followed by an evaluation of the pilot's workload.

Figure 18 presents the wavelet analysis results of the pilot's manipulation actions during the backward reconversion control strategy under benchmark performance index $J_{R1}$. The maximum energy input for the collective pitch stick at approximately 8 s significantly exceeds that of the longitudinal stick, but the high-frequency components are lower compared to those of the longitudinal stick. At 12 s, there is a lower-energy, high-frequency component, indicating a sense of urgency in the flight task, as indicated by Figure 4. The longitudinal stick, on the other hand, shows numerous small-amplitude, high-frequency components distributed vertically across different time nodes. This is caused by the continuous jumps in the longitudinal stick control rate within the constraint range, with some exceeding 4 rad/s. It is evident from Figure 4 that such control is challenging for the pilot.

According to the mapping relationship between frequency components and control loads proposed in Section 4 (Table 1), the corresponding HQR level for backward reconversion under benchmark performance index $J_{R1}$ indicates level 3, suggesting control difficulty for the pilot. Therefore, it is necessary to consider the pilot's control workload in the performance index to avoid significant manipulation of the collective pitch stick and the presence of high-frequency components in both the collective pitch stick and longitudinal stick controls.

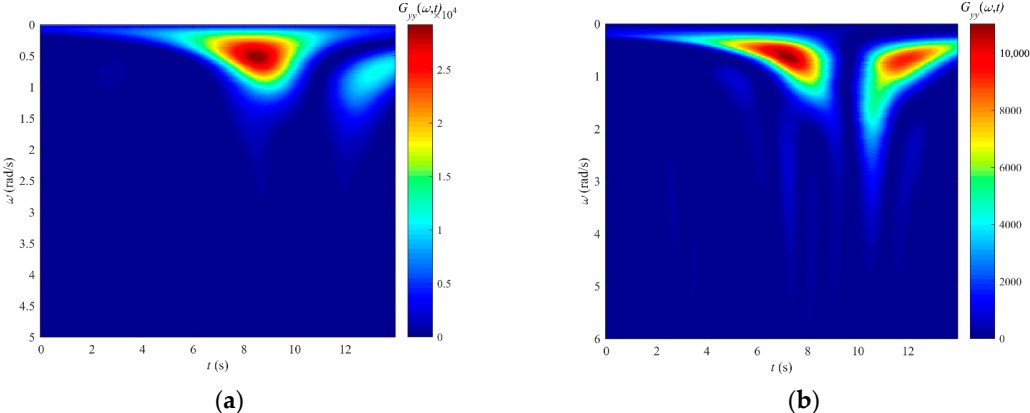

**Figure 18.** Wavelet analysis results of pilot manipulation under benchmark performance index $J_{R1}$. (**a**) Collective pitch stick wavelet transform result (unit: %2/(rad/s)). (**b**) Longitudinal stick wavelet transform result (unit: %2/(rad/s)).

### 6.3. Weighting of Pilot Workload in Performance Index

To incorporate the pilot workload into the performance index, this section introduces the weighted pilot control rate term based on benchmark performance index $J_{R1}$. This approach is consistent with that reported in Section 5.3.

$$
\begin{aligned}
\min J_{R2} = w_t \left( \tau_f - \tau_0 \right) + \\
\frac{w_p}{\tau_f - \tau_0} \int_{\tau_0}^{\tau_f} \left[ w_{col} \cdot u_{col}^2 / u_{col,\max}^2 + w_{lon} \cdot u_{lon}^2 / u_{lon,\max}^2 + w_n \cdot u_n^2 / u_{n,\max}^2 \right] d\tau
\end{aligned}
\tag{29}
$$

The weight coefficients are set as $w_{lon} = 0.5$, $w_{col} = 0.35$, and $w_n = 0.15$. The allocations of the pilot control workload weighting coefficient ($w_p$) and the time weighting coefficient ($w_t$) also significantly impact the control strategy for the backward reconversion procedure. To determine a reasonable allocation of weighting coefficients, in this section, we compare and analyze the pilot control strategy and flight state variations in association with different values of $w_p/w_t$. The goal is to determine the allocation of weighting coefficients for each parameter under performance metric $J_{R2}$. For analysis of the backward conversion procedure, notable values of 0, 3/7, 1/1, and 7/3 were selected to highlight the impact of $w_p/w_t$ on the control strategy.

Figure 19 shows the pilot control strategy and flight state under different $w_p/w_t$ values. It can be concluded that when the ratio of $w_p/w_t$ is 1/1 for the reconversion procedure, pilot manipulation is smooth, the height change is small, and the tilting process of the engine nacelle can be basically realized by the constant-angular-rate automatic tilting system (stabilized at 7.5°/s).

Therefore, the performance index is set as

$$
\begin{aligned}
\min J_{R2} = 0.5 \left( \tau_f - \tau_0 \right) + \\
\frac{0.5}{\tau_f - \tau_0} \int_{\tau_0}^{\tau_f} \left[ w_{col} \cdot u_{col}^2 / u_{col,\max}^2 + w_{lon} \cdot u_{lon}^2 / u_{lon,\max}^2 + w_n \cdot u_n^2 / u_{n,\max}^2 \right] d\tau
\end{aligned}
\tag{30}
$$

Figure 20 presents the wavelet analysis results of the pilot's manipulation actions during backward reconversion under benchmark performance index $J_{R2}$. Table 3 shows a comparison of wavelet analysis results between Figures 18 and 20.

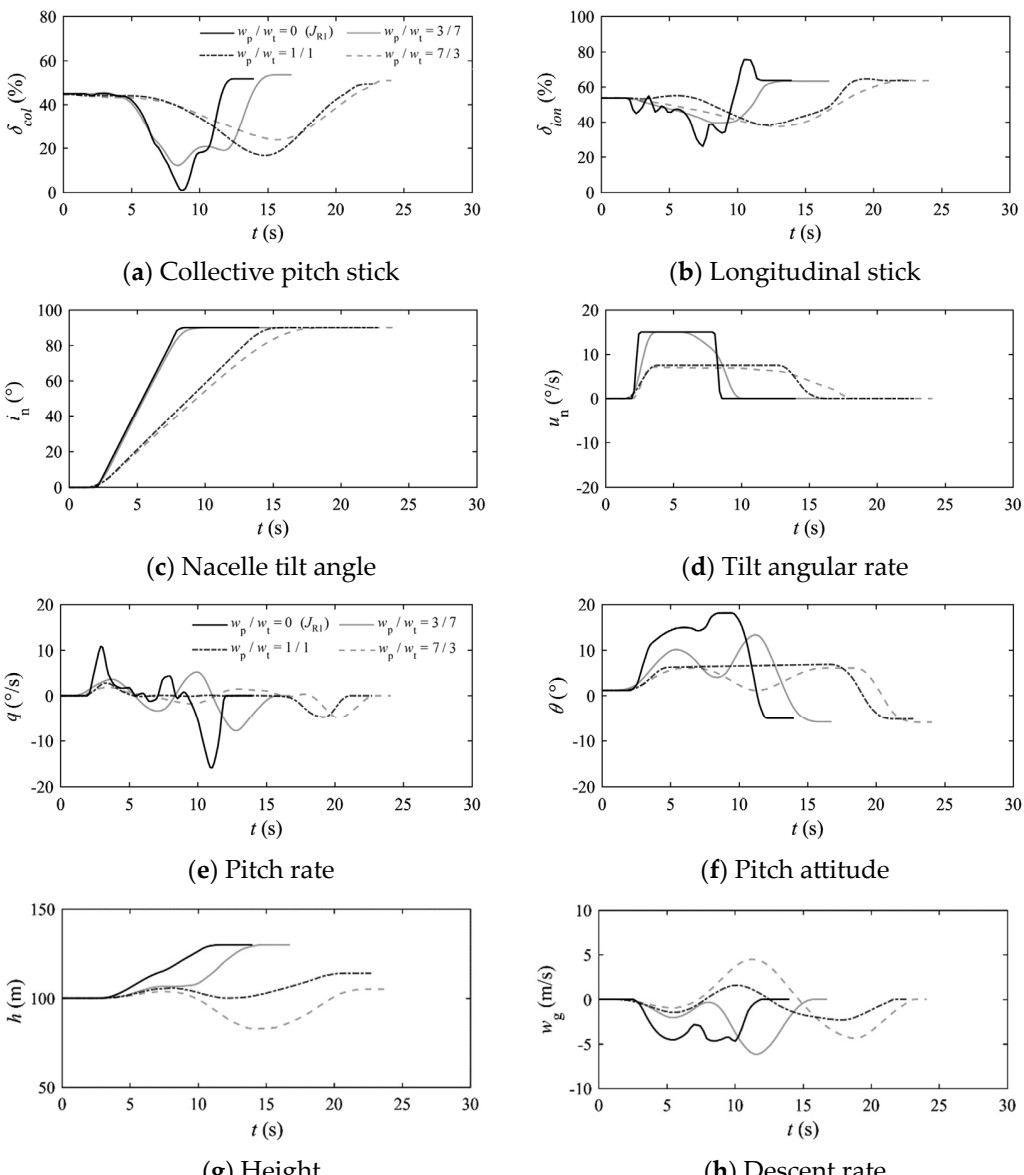

**Figure 19.** Backward reconversion in association with different $w_p/w_t$ values under performance index $J_{R2}$.

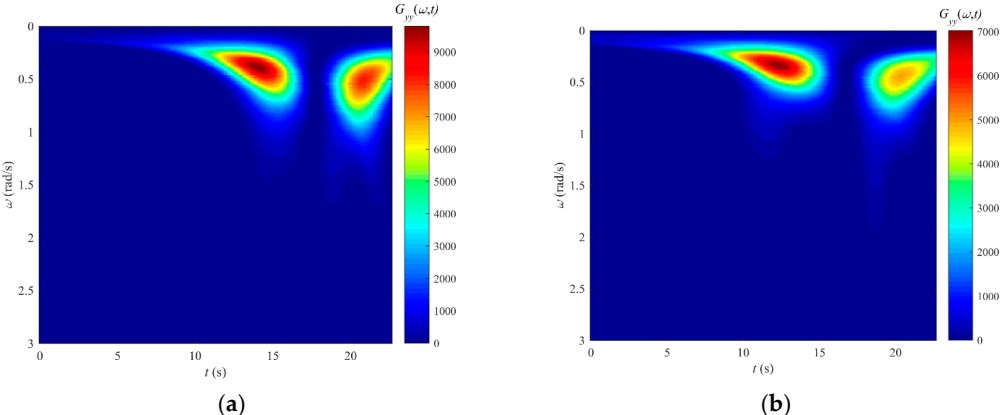

**Figure 20.** Wavelet analysis results of pilot manipulation under performance index $J_{R2}$. (**a**) Collective pitch stick wavelet transform result (unit: %2/(rad/s). (**b**) Longitudinal stick wavelet transform result (unit: %2/(rad/s)).

**Table 3.** Comparison of wavelet analysis results: Figure 18 vs. Figure 20, backward reconversion procedure.

| Control | Item | Figure 18 | Figure 20 |
|---|---|---|---|
| Collective pitch stick | Dominant frequency components | 0.1~2.0 rad/s | 0.2~0.8 rad/s, <1.7 rad/s |
| | Maximum energy | 29,000$\%^2$/(rad/s) | 9800$\%^2$/(rad/s) |
| Longitudinal stick | Dominant frequency components | 0.2~4 rad/s | 0.2~0.8 rad/s, <2.0 rad/s |
| | Maximum energy | 11,000$\%^2$/(rad/s) | 7000$\%^2$/(rad/s) |

Figure 20a shows the wavelet transform result of the collective pitch stick. It can be observed that the overall energy distribution of the collective pitch stick is similar to that shown in Figure 18a, but the maximum energy input is significantly reduced by 66%, as shown in Table 3. The main energy inputs are concentrated around 0.2 rad/s to 0.8 rad/s, and the low-energy (small-amplitude), high-frequency components are also less than 1.7 rad/s. Figure 20b displays the wavelet transform result of the longitudinal stick. A comparison with Figure 18b and Table 3 reveals a 36% decrease in the maximum energy of the longitudinal stick and a significant reduction in high-frequency components. The low-energy (small-amplitude), high-frequency components that appear after 12 s and 18 s are also less than 2 rad/s.

With reference to Table 1 in Section 4, it can be concluded that pilot manipulation mainly involves trim and flight path adjustments (with most of the energy inputs corresponding to frequency components below 0.8 rad/s). There is only a small degree of manipulation associated with transport aircraft maneuvers (a negligible portion of energy inputs corresponding to frequency components between 0.8 and 2.0 rad/s). According to the pilot control workload assessment method, it can be inferred that the pilot control workload under performance index $J_{R2}$ corresponds to level 1~2 (HQR 3~4). This indicates that backward reconversion a low-workload process with relatively simple and easy manipulation for the pilot.

## 7. Conclusions

In this study, we investigated control strategies and workload of tilt-rotor aircraft dynamic conversion and reconversion procedures. A nonlinear flight dynamics model of tilt-rotor aircraft with full flight modes was established. On this basis, a nonlinear optimal control model of dynamic conversion was constructed. An analytical method based on wavelet transform was proposed, which examines the mapping relationship between the amplitude of pilot control input, constituent frequencies, and control tasks. The investigations yielded the following conclusions.

(1) When the performance index considers the weight coefficient of pilot workload, the magnitude of collective stick inputs and longitudinal stick inputs are significantly reduced, promoting smoother changes in height and pitch attitude. Additionally, tilting of the engine nacelle can be accomplished at a lower and fixed angular rate. However, the pilot workload weight coefficient should not be too dominant; otherwise, overly smooth collective and longitudinal stick manipulation leads to a decrease in height.

(2) By incorporating the nonlinear optimal control model and the pilot workload evaluation method, in this study, we propose control strategies for both forward conversion and backward reconversion with low pilot workload. Through the utilization of appropriate performance indices, it is estimated that both the conversion and reconversion procedures exhibit low workload levels (HQR 3~4, level 1~2), making them relatively straightforward and easily manageable for the pilot.

(3) The approach of combining optimal control theory with TFR analysis methods offers a promising avenue for enhancing the development and assessment of pilot control strategies and workload. It is possible to evaluate and quantify a pilot's optimal control

strategy and workload early on, providing valuable insights for further optimization and refinement of the control system design.

## 8. Future Works

In order to further enhance the numerical simulation accuracy of dynamic conversion processes and improve the accuracy of pilot workload evaluation, further development and refinement of the pilot workload evaluation method are needed. The mapping relationship between the pilot's control amplitudes, frequency components, and pilot workload requires further validation and adjustment through flight experiments and simulations. Additionally, it would be beneficial to develop a pilot model that accurately reflects the pilot's workload, enabling a more precise evaluation of pilot workload during flight simulations.

Currently, the most effective and flexible approach to solve the optimal control problem of tilt-rotor aircraft conversion procedures involves transforming it into a nonlinear programming (NLP) problem using collocation methods, which can then be solved using nonlinear programming algorithms. However, this method involves significant computational complexity and cannot be performed in real time at present. One of our future goals is to achieve real-time computation on the flight control computer based on the current aircraft state, providing optimal control strategies for the entire crew, including the pilot, for different flight missions and environments.

**Author Contributions:** Conceptualization, X.Y. and R.C.; methodology, X.Y.; software, X.Y.; validation, X.Y., R.C. and Y.Y.; formal analysis, Y.Y.; investigation, X.Y.; resources, R.C.; data curation, X.Y.; writing—original draft preparation, X.Y.; writing—review and editing, X.Y. and Y.Y.; supervision, Y.Y.; project administration, X.Y. and R.C.; funding acquisition, X.Y. All authors have read and agreed to the published version of the manuscript.

**Funding:** This research is supported by the National Natural Science Foundation of China (No. NSFC-12202406) and the Natural Science Foundation of China (No. 11672128).

**Data Availability Statement:** The data that support the findings of this study are available from the corresponding author upon reasonable request.

**Conflicts of Interest:** The authors declare no conflict of interest.

## Abbreviations

List of abbreviations

| | |
|---|---|
| NOCP | Nonlinear optimal control problem |
| HQR | Handling quality rating |
| PIO | Pilot-induced oscillation |
| TFRs | Time-frequency representations |
| GTRS | Generic tilt-rotor aircraft simulation |
| NLP | Nonlinear programming |
| SQP | Series quadratic programming |
| STFT | Short-time Fourier transform |
| PSD | Power spectral density |
| db | Daubechies |
| MTE | Mission task element |

List of symbols

| | |
|---|---|
| $S_{wfs}$ | Wing area in the free stream |
| $S_{wss}$ | Wing area in the rotor slipstream |
| $R_w$ | Contracted wake radius |
| $\boldsymbol{x}_B, \boldsymbol{x}_F, \boldsymbol{x}_I$ | Fuselage state, rotor state, inflow state |
| $\delta_{col}, \delta_{lon}, \delta_{lat}, \delta_{ped}, i_n$ | Collective stick input, lateral stick input, longitudinal stick input, pedal input, nacelle tilting angle |

| | |
|---|---|
| $u, v, w$ | Linear velocities of the aircraft body axis system |
| $p, q, r$ | Angular velocities of the body axis system |
| $\phi, \theta, \psi$ | Roll, pitch, and yaw angles, respectively |
| $x, y, h$ | Position of the aircraft in the Earth axis system |
| $a_0, a_1, b_1$ | Taper, rear, and side angles of the rotor disk, respectively |
| $\nu_0, \nu_{1c}, \nu_{1s}$ | Non-dimensional terms for rotor uniform inflow, first-order cosine inflow, and first-order sine inflow, respectively |
| $H_p(s)$ | Perception system and neuromuscular system |
| $T_L, T_I, T_N, \tau_p$ | Feedforward time constant, lag time constant, neuromuscular response time, and delay time, respectively |
| $\boldsymbol{u}_d, \boldsymbol{u}_f$ | Delayed command inputs and final control input, respectively |
| $\boldsymbol{x}_p$ | Relevant state variables of the neuromuscular system |
| $\boldsymbol{A}, \boldsymbol{B}, \boldsymbol{C}, \boldsymbol{D}$ | Matrices of the pilot model in the state-space form |
| $u_c, u_{lon}, u_{lat}, u_{ped}, u_n$ | Control rates |
| $\alpha_W$ | Wing angle of attack |
| $P_r, P_n$ | Required and rated power output of the engine (two), respectively |
| $\Omega_0$ | Standard main rotor rotational speed |
| $k_1 \sim k_4$ | Constant scaling factors |
| $n_c, \omega_c$ | Number of cycles and frequency, respectively |
| $\psi(t)$ | Wavelet function |
| $G_{\delta\delta}(\omega, t)$ | Weighted power |
| $W_y(u, s)$ | Wavelet coefficient |
| $J_{C1}, J_{C2}$ | Performance indices for forward conversion |
| $J_{R1}, J_{R2}$ | Performance indices for backward reconversion |
| $w_t, w_p$ | Time weighting coefficient and pilot control workload weighting coefficient, respectively |
| $w_{col}, w_{lon}, w_n$ | Weighting coefficients for the corresponding control rates |

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
