# Peer review of "Research on Pilot Control Strategy and Workload for Tilt-Rotor Aircraft Conversion Procedure"

_aerospace, doi:10.3390/aerospace10090742_

Round 1

Reviewer 1 Report

This paper studies the pilot control strategy and workload of tilt-rotor aircraft dynamic conversion procedure between helicopter mode and fixed-wing mode. Based on the nonlinear flight dynamics model, a nonlinear optimal control model of dynamic conversion is constructed, considering factors such as conversion corridor limitations, pilot control, flight attitude, engine rated power, and wing stall effects. To assess pilot workload, an analytical method based on wavelet transform is proposed. In general, this work is of great significance in terms of tilt-rotor aircraft design and operation. I have several main concerns as follows:

1).What the authors are talking about using "nonlinear optimal control model"? Whether or not a real human pilot is in the loop to operate the tilt-rotor aircraft?Or, do you mean: the "nonlinear optimal control model" is a representative of the human pilot?The authors should give the control structure blocks (charts) including the aircraft, control model for the simulation closed-loop model. More details need to be given to make this clear.

2).In the paragraph "Therefore, this paper first employs nonlinear optimal control theory to ...", the innovations are not clear, and need to be reformulated, e.g., what is the difference or the advantages of your methods.

3). what is the application places for the findings (control strategies) in real flight of tilt-rotor aircraft? Is a real human pilot needed or not in future is your control model is adopted in flight control computer?

English writing is good in general, however, in some places grammar should be checked for this paper. 

Reviewer 2 Report

Please, check the uploaded file.

Round 2

Reviewer 1 Report

The author has well replied the reviewer's comments.This paper can be accepted after minor English grammar revisions.

Minor English grammar revisions are still needed. The authors should read through the manuscript, and make possible changes.

Reviewer 2 Report

Thank you for your valuable answers, I have appreciated the effort to answer all the questions related to the manuscript.